# A De Novo Optimized Cell-Free System for the Expression of Soluble and Active Human Tumor Necrosis Factor-Alpha

**DOI:** 10.3390/biology11020157

**Published:** 2022-01-19

**Authors:** Nawal Abd El-Baky, Esmail M. EL-Fakharany, Soraya A. Sabry, Ehab R. El-Helow, Elrashdy Mustafa Redwan, Amira Sabry

**Affiliations:** 1Therapeutic and Protective Proteins Laboratory, Protein Research Department, Genetic Engineering and Biotechnology Research Institute (GEBRI), City of Scientific Research and Technological Applications (SRTA-City), New Borg El-Arab City, Alexandria P.O. Box 21934, Egypt; eelfakharany@srtacity.sci.eg (E.M.E.-F.); asabry@srtacity.sci.eg (A.S.); 2Department of Botany and Microbiology, Faculty of Science, Alexandria University, Alexandria P.O. Box 21568, Egypt; sorayasabry@alexu.edu.eg (S.A.S.); elhelow@alexu.edu.eg (E.R.E.-H.); 3Biological Sciences Department, Faculty of Science, King Abdulaziz University, Jeddah P.O. Box 80203, Saudi Arabia; lradwan@kau.edu.sa

**Keywords:** anticancer, human tumor necrosis factor-alpha, in vitro protein synthesis, optimization, response surface methodology

## Abstract

**Simple Summary:**

As a result of increasing demand for the pleiotropic cytokine TNF-α, recombinant human TNF-α protein with appropriate bioactivities was produced in several heterologous in vivo expression systems. While in vivo expression of this cytokine is laborious and lengthy, cell-free or in vitro expression system has the benefits of speed, simplicity, flexibility, focus of all the system energy on target protein synthesis alone, besides high soluble and functional protein yield. Therefore, we employed and optimized an *E. coli*-based cell-free system for the first time to express recombinant human TNF-α. Our findings revealed that cell-free expression system can be an alternative platform for producing soluble and functionally active recombinant TNF-α with a yield of 390 µg/mL in only 2 h at a temperature of 40 °C for further research and clinical trials.

**Abstract:**

Cell-free (in vitro) expression is a robust alternative platform to the cell-based (in vivo) system for recombinant protein production. Tumor necrosis factor-alpha (TNF-α) is an effective pro-inflammatory cytokine with pleiotropic effects. The aim of the current study was de novo optimized expression of soluble and active human TNF-α by an in vitro method in an *E. coli*-based cell-free protein synthesis (CFPS) system and its biological activity evaluation. The codon-optimized synthetic human TNF-α gene was constructed by a two-step PCR, cloned into pET101/D-TOPO vector and then expressed by the *E. coli* CFPS system. Cell-free expression of the soluble protein was optimized using a response surface methodology (RSM). The anticancer activity of purified human TNF-α was assessed against three human cancer cell lines: Caco-2, HepG-2 and MCF-7. Data from RSM revealed that the lowest value (7.2 µg/mL) of cell-free production of recombinant human TNF-α (rhTNF-α) was obtained at a certain incubation time (6 h) and incubation temperature (20 °C), while the highest value (350 µg/mL) was recorded at 4 h and 35 °C. This rhTNF-α showed a significant anticancer potency. Our findings suggest a cell-free expression system as an alternative platform for producing soluble and functionally active recombinant TNF-α for further research and clinical trials.

## 1. Introduction

Tumor necrosis factor-alpha is a pleiotropic cytokine with a pro-inflammatory and immunoregulatory function [1,2]. It is involved in host defense against microbial infections, control of cell survival and proliferation and efficient antitumor activities. It is also called cachectin, endotoxin-induced factor in serum, or differentiation-inducing factor [3,4,5]. This cytokine is secreted from activated macrophages, monocytes and natural killer (NK) cells in response to any immunological challenge [2,6,7,8]. Human TNF-α is encoded by a single copy gene located in chromosome 6, which has four exons and three introns and the protein is mostly encoded in the last exon. Our previous analysis for the sequence of TNF-α from different species revealed sequence homology with a percentage that ranged from 99.57% (between human and chimpanzee TNF-α proteins) to 22.33% (between frogs and fish TNF-α proteins). Interestingly, we found that the intrinsic disorder profile of this family of proteins is characterized by a remarkable similarity and subsequently are evolutionarily conserved [9].

Mammalian TNF-α exists as a non-glycosylated type II transmembrane precursor protein (membrane-bound TNF-α or mTNF-α) with a molecular mass of about 26 kDa (157 amino acids in addition to a leader sequence of 76 amino acids) [3,8,9]. The mTNF-α is subjected to proteolytic cleavage by the metalloprotease TNF-α converting enzyme (TACE or ADAM17) at alanine-66 and valine-67 leading to the formation of free soluble homotrimer TNF-α (sTNF-α) of about 17 kDa molecular weight. Both mTNF-α and sTNF-α can bind to TNF-α receptors (TNFR1 or TNFR2) and exhibit pleiotropic effects on various cell types [10]. However, the sTNF-α protein exerts in vitro and in vivo cytotoxic activity against particular tumor cells via binding to and activating TNFR1 [11,12]. This killing TNFR1 signaling pathway makes sTNF-α protein (mature TNF-α) a promising drug for cancer treatment in clinical trials and market.

As a result of increasing demand for sTNF-α protein, recombinant human TNF-α protein with appropriate bioactivities was produced in several heterologous protein expression systems such as *Escherichia coli* [13,14], *Lactococcus lactis* [15], *Streptomyces lividans* [16], a baculovirus expression system [17,18], or mouse embryonic fibroblasts [19]. Continuous efforts have been made to maximize TNF-α productivity from an *E. coli* expression system, including gene codon optimization, choice of the strongest expression promoters and optimization of culture and induction conditions to obtain more soluble recombinant protein over inclusion bodies [13,14,20,21,22]. Eukaryotic expression systems, as in the case of baculovirus expression systems and mammalian cell expression systems offer the unique advantage of soluble protein production without lengthy and laborious optimization of conditions of culture and induction, which is essential in *E. coli* systems, but have the drawbacks of high cost of cell culture and maintenance of the laboratory [23,24,25].

Cell-free or in vitro protein synthesis employs the cellular protein machinery to synthesize proteins directly outside the living host cells, without any constraints of cell membrane [26,27,28]. In cell-free systems, proteins are synthesized in a crude cell extract. Thus, CFPS platforms have the benefit of avoiding typical steps for cell-based protein synthesis such as transformation, clone selection, expression induction, optimization of culture and induction conditions and cell lysis, in addition to inclusion bodies’ denaturation before purification and refolding of purified protein. CFPS platforms were derived from eukaryotes (e.g., rabbit reticulocytes, wheat germ, human cells, or insects) or prokaryotes (e.g., *E. coli*, *Vibrio natriegens*, or *Bacillus subtilis*). The choice of extract source depends predominantly on the origin of the target protein, its complexity and further downstream applications [26,28]. The components of CFPS reaction mixture are template DNA encoding the target protein, cell lysate and supplements to sustain in vitro transcription and translation [27,29,30,31].

CFPS systems are becoming increasingly used as an alternative to in vivo systems because of the relatively high reaction speed, simplicity and flexibility; focus of all the system energy on target protein synthesis alone; easy and direct manipulation of CFPS open reaction; high soluble and functional protein yield and the possibility of using PCR fragments or mRNA directly as a template without cloning [26,27,28,32]. Furthermore, the in vitro approach is a salvation for the synthesis of toxic proteins for the metabolism of the host cells [33]. Additionally, CFPS system permits incorporation of non-natural or chemically modified amino acids at specific positions during translation, which generates novel proteins [26]. However, the price of CFPS systems is relatively high [32,34]. 

The current work is the first report where the cell-free approach was employed to produce rhTNF-α protein. After optimizing cell-free expression of soluble protein using RSM, the anticancer activity of purified protein was investigated against Caco-2, HepG-2 and MCF-7. The cell-free synthesized human TNF-α demonstrated a cytotoxic effect in a concentration-dependent manner on test cancer cells. This study recommends the use of CFPS as an alternative platform for the synthesis of soluble and functionally active TNF-α products for further academic research and clinical trials.

## 2. Materials and Methods

### 2.1. Bacterial Strains and Vectors

*E. coli* XL2-Blue MRF’ ultracompetent cells kit used for gene cloning was purchased from Stratagene (Heidelberg, Germany).

TOPO TA cloning kit (Invitrogen, Carlsbad, CA, USA) was used for gene cloning into pCR 2.1-TOPO cloning plasmid.

A champion pET Directional TOPO expression kit was obtained from Invitrogen and was used for cell-free protein expression of human TNF-α under control of T7 promoter in pET101/D-TOPO expression plasmid.

### 2.2. Rare Codon Analysis of the DNA Coding Sequence for Human TNF-α

First, the amino acid sequence of human sTNF-α was obtained from the Universal Protein Resource (UniProt) website, https://www.uniprot.org/ (accessed on 1 February 2017); subsequently, it was reverse translated using the following website: http://www.molbiol.ru/eng/scripts/01_19.html (accessed on 1 February 2017). The resulted DNA coding sequence was tested for existence of rare codons using GenScript Rare Codon Analysis Tool (https://www.genscript.com/tools/rare-codon-analysis (accessed on February 2017)) to confirm its successful expression in the *E. coli* extract-based CFPS system.

### 2.3. Codon Optimization of the DNA Coding Sequence for Human TNF-α

The BioEdit program version 7 was utilized to reverse translate the amino acid sequence of human sTNF-α based on a codon usage table generated from 681 *E. coli* genes found in GenBank release 63, to obtain a DNA sequence with high-level expression by protein synthesis machinery of the *E. coli* CFPS system. 

### 2.4. Restriction Map of Human TNF-α Gene

To design cloning experiments and analyze the positive clones, a gene restriction map was generated using the Sequence Extractor website: https://www.bioinformatics.org/seqext/ (accessed on 1 March 2017). This map identifies the restriction enzymes sites in the human TNF-α gene.

### 2.5. Primers Design for the Synthesis of Codon-Optimized Human TNF-α Gene

#### 2.5.1. Oligonucleotide Primers for Gene Synthesis

To synthesize the codon-optimized human TNF-α gene, 20 oligonucleotides (10 forward and 10 reverse), each with average length of 40 nucleotides were designed to span most of both strands of the gene sequence as previously described [35,36]. Oligonucleotides were chemically synthesized at Invitrogen and purified using the desalting method. The designed oligonucleotides overlap neighboring oligonucleotides by 13–21 bp and their rare codons were replaced by abundant ones.

#### 2.5.2. Gene-Specific Primers for Amplification and Subsequent Cloning into Vectors

For amplification and subsequent cloning into pCR 2.1-TOPO plasmid, the forward PCR primer length was 22 nucleotides. It was designed to contain the *BamH*I restriction site (GGATCC) followed by an initiation ATG codon. Meanwhile, for cloning into pET101/D-TOPO, the forward PCR primer length was 22 nucleotides and contains 5′ CACC overhang followed by the *BamH*I restriction site and ATG codon. 

The reverse primer designed for cloning into pCR 2.1-TOPO and pET101/D-TOPO vectors contains an *Xho*I restriction site (CTCGAG) and has a length of 21 nucleotides. 

### 2.6. Two-Step PCR-Mediated Gene Construction

#### 2.6.1. PCR Assembling Step

The human TNF-α gene was assembled by PCR using the designed overlapped oligonucleotide primers and BioReady *Pfu* DNA polymerase with proofreading properties (Hangzhou Bioer Technology Co., Hangzhou, China). 

Working on ice, the following reagents were added to a sterile DNase- and RNase-free PCR tube as follows: twenty oligonucleotide primers (5 pM/each), *Pfu* reaction buffer with 1.5 mM MgCl_2_ (1 X), dNTP Mix (0.5 mM) and BioReady *Pfu* DNA polymerase (2.5 U). The reaction was completed to 25 µL with sterile water for injection and mixed gently by tapping PCR tube.

The cycling conditions were as follows: 35 cycles of 94 °C for 30 s, 58 °C for 2 min, 72 °C for 2 min, followed by a final extension of 72 °C for 10 min [36]. 

#### 2.6.2. PCR Amplification Step

Working on ice, in a sterile DNase- and RNase-free PCR tube, the following components were mixed as follows: outermost oligonucleotide primers (20 pM/each), template (an assembled product from PCR assembling step, 1 µL), *Pfu* reaction buffer with 1.5 mM MgCl_2_ (1 X), dNTP Mix (0.5 mM) and BioReady *Pfu* DNA polymerase (2.5 U). The reaction was completed to 25 µL with sterile water for injection and mixed gently. 

The cycling conditions were as follows: 95 °C for 3 min followed by 35 cycles of 94 °C for 30 s, different annealing temperatures (57, 57.5, 58, 58.5 and 59 °C) for 2 min, 72 °C for 2 min and final extension of 72 °C for 10 min [36].

Both the assembly reaction product and the final amplified full-length gene were analyzed by 2% agarose gel electrophoresis.

### 2.7. Cloning of Human TNF-α Gene 

*Pfu* DNA polymerase only produces a blunt-ended PCR product and removes the 3′ adenine overhang required for TA cloning into pCR 2.1-TOPO. Accordingly, the blunt-ended PCR product of the human TNF-α gene amplified at an annealing temperature of 58 °C was reamplified using gene-specific forward and reverse primers designed for subsequent cloning into pCR 2.1-TOPO and Dream *Taq* DNA polymerase obtained from Thermo Fisher Scientific (Bedford, MA, USA) (2.5 U/25 µL reaction), which is capable of adding a single 3′ adenine overhang. Subsequently, the PCR was followed by 2% agarose gel electrophoresis and the human TNF-α gene was extracted and purified from 2% agarose gel using the QIAquick gel extraction kit (Qiagen, Hilden, Germany) according to the manufacturer’s instructions.

The purified human TNF-α amplicon was inserted into pCR 2.1-TOPO cloning vector using a TOPO TA cloning kit following the company’s instruction manual and transformed into *E. coli* XL2-Blue MRF’ ultracompetent cells. The positive clones were confirmed by restriction digestion and nucleotide sequencing.

### 2.8. Construction of Expression Plasmid for CFPS

The gene fragment resulted from restriction cutting of the pCR 2.1-TOPO cloning vector carrying the human TNF-α correct sequence was extracted from 1.2% agarose gel. Then, it was reamplified by PCR at annealing temperature of 58 °C using gene-specific forward and reverse primers designed for cloning into the pET101/D-TOPO vector and BioReady *Pfu* DNA polymerase (2.5 U/25 µL reaction) to produce the blunt-ended PCR product. The PCR product was inserted into pET-TOPO expression vector as recommended by the manufacturer to yield a pET-hTNF-α expression construct. The recombinant plasmid was transformed into *E. coli* XL2-Blue MRF’ ultracompetent cells as a cloning host for amplification and storage of the pET-hTNF-α expression construct. The positive clones were analyzed by PCR using gene-specific forward and reverse primers designed for cloning into the pET101/D-TOPO vector and by restriction digestion in addition to nucleotide sequencing to confirm the presence and correct orientation and sequence of the insert. 

### 2.9. Cell-Free Synthesis of Recombinant Human TNF-α Protein

Cell-free (In Vitro) expression of the rhTNF-α protein from the pET-hTNF-α construct was carried out using the rapid translation system RTS 100 *E. coli* HY kit (CFPS system derived from *E. coli* crude cell extract) obtained from 5 PRIME (Hamburg, Germany) as previously reported [37]. According to the instruction manual, the reagents and working solutions were kept on ice and were added into a sterile eppendorf tube in the following order: 12 µL *E. coli* S30 lysate, 10 µL reaction mix, 12 µL amino acids, 1 µL methionine, 5 µL reconstitution buffer and 10 µL of pET-hTNF-α construct at concentration of 0.5 µg dissolved in sterile water for injection. Additionally, a negative control reaction was performed using the same components except for plasmid construct. The reactions were mixed gently and incubated at 30 °C for 6 h. Afterwards, the reaction tubes were stored at −20 °C for SDS-PAGE analysis and ELISA of recombinant human TNF-α.

### 2.10. SDS-PAGE Analysis of Cell-Free Synthesized Human TNF-α Protein

The cell-free synthesized human TNF-α protein samples were prepared for SDS-PAGE according to the RTS 100 *E. coli* HY kit manual as follows: the soluble fraction of the cell-free synthesized human TNF-α protein was harvested by centrifugation for 5 min at 12.000 rpm and the protein was precipitated with the addition of 500 µL of −20 °C cold acetone in an Eppendorf tube, mixed and incubated on ice for 10 min. Then, it was centrifuged at 12.000 rpm for 15 min at 4 °C. The supernatant was discarded, the pellet was air dried for 15 min and resuspended in 50 µL of 1 X sample buffer. Afterwards, samples were boiled for 10 min, centrifuged and loaded onto the SDS gel.

### 2.11. Quantitative Estimation of Synthesized Human TNF-α by ELISA

The soluble fraction of the cell-free synthesized human TNF-α protein was harvested by centrifugation for 5 min at 12.000 rpm and the protein was precipitated with the addition of 500 µL of −20 °C cold acetone in an Eppendorf tube, mixed and incubated on ice for 10 min. Then, it was centrifuged at 12.000 rpm for 15 min at 4 °C. The supernatant was discarded, the pellet was air dried for 15 min and resuspended in PBS. Quantitative estimation of the soluble human TNF-α protein synthesized in the CFPS system was performed using a sandwich high-sensitivity human TNF alpha PicoKine ELISA kit (Boster Biological Technology, Wuhan, China) according to the manufacturer’s instructions. The developed yellow color was measured at 450 nm by a microplate reader (BMG Labtech, Ortenberg, Germany). The cell-free synthesized human TNF-α concentration was calculated from a calibration curve made by using different concentrations of the recombinant human TNF-α standard.

### 2.12. Two-Step Chromatographic Purification of Cell-Free Synthesized Human TNF-α

The first step was carried out using AKTAprime plus fast performance liquid chromatography (FPLC) protein separator system with an ion-exchange chromatography column packed with Q-Sepharose fast flow resin (strong anion exchanger), from GE Healthcare Life Sciences Products (Cardiff, UK). The soluble protein fraction of CFPS reaction of human TNF-α was dialyzed against buffer A (20 mM Tris-HCl, pH 8) at 4 °C for 24 h with stirring, using a dialysis bag with molecular weight cut-off 14–18 kDa.

During these 24 h, the dialysis buffer was replaced twice with fresh buffer. Directly after dialysis, the human TNF-α soluble fraction was loaded onto a Q-Sepharose fast flow column, which was previously equilibrated by buffer A. Then, the column was washed with the same buffer until the ultraviolet (UV) absorbance at 280 nm was stable around zero and eluted with a 0–100% linear gradient of buffer B (20 mM Tris-HCl, pH 8, 1 M NaCl) at a flow rate of 1 mL/min. Fractions were collected, respectively, and further analyzed by 12% SDS-PAGE. At the same time, the total protein concentration was measured by Bradford assay.

In the second step, the fraction containing the intended protein was pooled, then dialyzed against 0.01 M sodium phosphate buffer, pH 6 and the pH was carefully adjusted to 6. The processed fraction was applied to a heparin- Sepharose 6 fast flow affinity column equilibrated with the same buffer and the column was placed on rock and roller shaker at 4 °C for 2 h. Afterwards, the column was washed with 0.01 M sodium phosphate buffer, pH 6 to remove proteins that unspecifically to the resin and the intended protein was eluted by 0.01 M sodium phosphate buffer, pH 6 with a gradient concentration of 0.1–1 M NaCl. All fractions were collected and analyzed by 12% SDS-PAGE and Bradford assay. Percent recovery was calculated according to the following equation: recovery % = (protein concentration in the collected fraction(s)/protein concentration in CFPS reaction) × 100. Fractions containing cell-free synthesized human TNF-α protein were dialyzed against PBS, pH 7.2 and stored at −20 °C.

### 2.13. RSM for the Optimization of Cell-Free Expression of Soluble Human TNF-α

#### 2.13.1. Experimental Design

A response surface methodology was used to optimize the soluble human TNF-α protein expression in *E. coli* CFPS system. A 2^2^ fractional factorial central composite rotary design for two independent variables each at five levels was employed, indicating that 13 experiments were required as a whole. Soluble human TNF-α protein production was taken as the response and it was estimated using quantitative ELISA.

#### 2.13.2. Statistical Analysis

The experimental data derived from the design were analyzed by multiple regressions through the least square method to fit the following second-order polynomial equation [38]:*Y* = β*o* + ∑ β*ix**i* + ∑ β*ij x**ix**j* + ∑ β*iix**ii*^2^(1)
where *Y* is the measured response variable, β*o*, β*i*, β*ij* and β*ii* are constant and regression coefficient of the model, *x**i* and *x**j* represent the independent variables in coded values. The second-order polynomial coefficients were calculated via Minitab 14 software to evaluate the dependent variable response. Response surface as well as the contour plots were designed using the same software, to assess the interactions between the different variables. Analysis of variance (ANOVA) was performed so that the second-order polynomial equation fits for all response variables. The significance of the model equation and model terms were estimated by Fisher test (*F*-test) and its associated probability *P*(*F*), while the quality of fit of the polynomial model equation was expressed by the coefficient of determination (*R*^2^) and adjusted *R*^2^ (Adj 𝑅^2^). An optimization tool of Minitab 14 software was utilized, in which a combination of different optimized parameters that achieved the maximum response (soluble protein cell-free expression) was tested experimentally to confirm the model validity.

### 2.14. Bioassays of Cell-Free Synthesized Human TNF-α Protein

#### 2.14.1. Human Cancer Cell Lines

Human colon cancer cell line Caco-2 HTB-37, human hepatoma cell line HepG-2 HB-8065 and human breast cancer cell line MCF-7 HTB-22 were obtained from American Type Culture Collection (ATCC), USA.

#### 2.14.2. Cell Culture

The Caco-2 cell line was cultured in DMEM high glucose medium supplemented with 10% fetal bovine serum (FBS) and 2 mM L-glutamine, while the HepG-2 and MCF-7 cell lines were maintained in RPMI-1640 augmented with 10% FBS and 2 mM L-glutamine. All the mentioned cell lines were incubated at 37 °C, in a humidified 5% CO_2_ incubator (BINDER, Germany). The cells were passaged twice to thrice per week, enzymatically with 0.25% trypsin-EDTA (1 X) and subcultured in 25 cm^2^ tissue culture flasks. 

#### 2.14.3. Determination of the Cytotoxicity of Cell-Free Synthesized Human TNF-α against Normal Human Cells

Human peripheral blood mononuclear cells (PBMCs) were isolated according to the method described by Boyum (1968) [39]. Briefly, fresh heparinized blood was carefully layered over an equal volume of the Ficoll–Hypaque solution (density of 1.077 g/mL) and centrifuged at 2.000 rpm for 30 min at room temperature. The mononuclear cells, located at the interface between the plasma (upper layer), the Ficoll–Hypaque (bottom) was collected, washed by PBS and centrifuged at 1.650 rpm for 5 min at room temperature. The PBMC pellets were resuspended in RPMI-1640 with 10% FBS. The cell count and viability were assessed using trypan blue dye exclusion assay. 

The cytotoxic effect of cell-free synthesized human TNF-α on PBMCs was evaluated using 3-(4,5-dimethylthiazol-2-yl)-2,5-diphenyltetrazolium bromide (MTT) dye assay as previously described by Mosmann, 1983 [40] and El-Baky et al., 2011 [37]. Shortly, 1 × 10^5^ mononuclear cells/well were seeded in a 96-well cell culture plate. The cells were treated with serial concentrations of purified cell-free synthesized human TNF-α in RPMI-1640 complete medium and were incubated at 37 °C in 5% CO_2_ incubator for 72 h. Afterwards, 20 μL of 5 mg/mL MTT were added to each well and the plate was incubated at 37 °C for 4 h followed by centrifugation at 2.000 rpm for 10 min at room temperature. Then, the MTT solution was removed and 100 µL dimethyl sulfoxide (DMSO) were added to dissolve the formazan crystals which were produced by the viable cells. The absorbance was measured with a microplate reader at 570 nm. All the measurements were performed in duplicate and untreated cells were used as a control. Finally, the percentages of cell viability and cytotoxicity were calculated compared to the untreated cells. The values of half-maximal inhibitory concentration (IC_50_) and the safe dose (concentration that achieves 100% cell viability, EC_100_) of cell-free synthesized human TNF-α were estimated by Graphpad Instat software version 3.

#### 2.14.4. Determination of the Cytotoxicity of Cell-Free Synthesized Human TNF-α against Human Cancer Cell Lines

Anticancer effect of cell-free synthesized human TNF-α was assessed against Caco-2, HepG-2 and MCF-7. All cancer cell lines (5 × 10^3^ cells/well) were seeded in 96-well cell culture plates. After 24 h, serial dilutions of EC_100_ of human TNF-α that caused 100% PBMCs viability (to calculate EC_100_ values against cancer cell lines) or serial concentrations of purified cell-free synthesized human TNF-α (to calculate IC_50_ values against cancer cell lines), were incubated with the three cancer cell lines for 72 h at 37 °C in 5% CO_2_ incubator. MTT assay was performed and the percentages of cell viability and cytotoxicity were estimated. The IC_50_ and EC_100_ values against test cancer cell lines were calculated using the Graphpad Instat software version 3. 

#### 2.14.5. Assessment of Cancer Cell Death Using Nuclear Staining

Antitumor activity of cell-free synthesized human TNF-α against Caco-2, HepG-2 and MCF-7 cancer cell lines, was determined by staining cells with ethidium bromide (EtBr) and acridine orange (AO) dye mix [41]. All cancer cells (5 × 10^3^ cells/well) were seeded in 96-well cell culture plates. After 24 h, the cells were treated with the IC_50_ of human TNF-α and incubated for 72 h at 37 °C in 5% CO_2_ incubator. Subsequently, plates were centrifuged at 1.000 rpm for 5 min at room temperature and 16 µL of EtBr and AO dye solution (50 μg/mL) were added to each well containing 150 µL media. Afterwards, the plates were left for 1–2 min to allow the dye to diffuse into the cells and visualized under fluorescent phase-contrast microscope (Olympus, Tokyo, Japan). The EtBr and AO staining method was performed in duplicates and untreated cells were used as a control.

## 3. Results

### 3.1. Codon Optimization of the Human TNF-α Gene

To optimize the coding sequence for the human TNF-α gene for expression in the *E. coli* CFPS system, in silico tools were used. The amino acid sequence of human sTNF-α containing 157 residues (amino acids: 77–233) (Uniprot identifier: P01375) (Figure 1) was obtained, reverse translated and tested for rare codons using the GenScript rare codon analysis tool. As shown in Table 1, the codon adaptation index (CAI) of human TNF-α gene is 0.7. A CAI of 1.0 is considered ideal for expression in *E. coli* and, thus, a CAI of 0.7 predicts that the human TNF-α gene will be expressed poorly by protein synthesis machinery of the *E. coli* CFPS system. Furthermore, the percentage of low frequency (<30%) codons based on *E. coli* is 12%, revealing that human TNF-α gene employs tandem rare codons that may reduce the translation efficiency or even disengage the machinery of translation. In view of that, by using BioEdit, the gene was codon-optimized to allow its heterologous production in the *E. coli* CFPS system without affecting the functional amino acid sequence.

### 3.2. Synthesis of Full-Length of Codon-Optimized Human TNF-α Gene Using Assembly and Amplification PCR Strategy

The restriction site analysis of the codon-optimized human TNF-α coding sequence is shown in Figure 2, presenting the relative positions of the restriction enzyme sites. It revealed that the gene does not comprise sites for *BamH*I and *Xho*I; thus, they were used in designing gene-specific primers for subsequent cloning experiments. To synthesize the codon-optimized human TNF-α gene, oligonucleotides encoding both strands of the DNA duplex were designed as illustrated in Figure 3, while gene-specific primers were constructed as depicted in Table 2.

The full-length of the codon-optimized human TNF-α gene (474 bp) was synthesized by a two-step gene synthesis procedure—assembly (Figure 4A) and amplification PCR (Figure 4B). In assembly PCR, all the designed overlapped oligonucleotides (10 forward and 10 reverse, which have average length of 40 nucleotides and span most of both strands of the gene sequence) were mixed at equivalent molar concentrations in the same reaction to assemble short DNA duplexes, thus priming the elongation by the proofreading DNA polymerase. These short duplexes were used in amplification PCR as substrate for formation of longer duplexes, finally resulting in the synthesis of the full-length gene, which was amplified using outermost oligonucleotide primers at concentration of 20 pM/each. The effect of different annealing temperatures was investigated (Figure 4B) and revealed that the purity and yield of the human TNF-α gene were slightly improved at the temperature 58 °C.

### 3.3. Cloning of Human TNF-α Gene into Cloning and Expression Vectors

The synthesized codon-optimized human TNF-α gene was cloned into pCR 2.1-TOPO and transformed into *E. coli* XL2-Blue MRF ultracompetent cells. Subsequently, the positive clones were subjected to restriction digestion using the *EcoR*I enzyme (Figure 5). The produced bands of restriction digestion were at 3881 bp and 505 bp, which matched the sizes of vector fragment and insert, respectively. Sequencing results confirmed that a positive clone carrying recombinant pCR 2.1-TOPO-human TNF-α plasmid with the correct human TNF-α sequence was obtained (data not shown). Eventually, the gene fragment from the pCR 2.1-TOPO-human TNF-α plasmid was subcloned into the pET101/D-TOPO expression vector and transformed into *E. coli* XL2-Blue MRF cells. The presence and correct orientation of the human TNF-α gene in the vector were verified by PCR (Figure 6A) and restriction digestion (Figure 6B). The results on the gel were consistent with the predicted sizes of the bands.

### 3.4. Cell-Free Synthesis of Recombinant Human TNF-α Protein

The recombinant human TNF-α protein expressed in RTS 100 CFPS system derived from *E. coli* S30 crude cell extract was analyzed by SDS-PAGE. A highly expressed protein band of approximately 21 kDa was separated on SDS-PAGE (Figure 7), which is consistent with the theoretically expected molecular weight calculated by the ExPASy ProtParam online tool for cell-free expressed human TNF-α protein along with added tags from the expression vector. Furthermore, this band was absent in case of the negative control reaction.

Quantitative estimation of the amount of soluble rhTNF-α protein produced by RTS 100 CFPS system was performed using a sandwich highly sensitive ELISA kit. The results revealed that the concentration of cell-free synthesized human TNF-α protein was 200 µg/mL without expression optimization. 

### 3.5. Two-Step Chromatographic Purification of Cell-Free Synthesized Human TNF-α Protein

The soluble protein fraction of the CFPS reaction of human TNF-α was purified using a two-step chromatographic strategy, namely ion exchange (anion exchange) and heparin–Sepharose affinity chromatography. In the first step, human TNF-α protein was eluted by approximately 0.5 M sodium chloride (50% buffer B) at about 64 min (Figure 8A), with a recovery of 49.98% (Table 3). The total protein concentration was estimated using Bradford assay and found to be 160.3 µg/mL (Table 3). In the second step, the intended protein was eluted by 0.2 M sodium chloride. The recovery percentage and total protein concentration were 21.98% and 70.5 µg/mL, respectively (Table 3). The purity of the eluted fractions obtained by the two-step chromatographic purification was confirmed by SDS-PAGE analysis (Figure 8B). Based on these results, it can be presumed that the cell-free synthesized human TNF-α protein was purified to near homogeneity using ion exchange (anion exchange) and heparin–Sepharose affinity chromatography.

### 3.6. RSM for the Optimization of Cell-Free Synthesis of Human TNF-α Protein

The statistical approach of RSM (many factors at a time approach) can optimize independent variables to maximize the response (the cell-free production of soluble and active human TNF-α). The central composite design (CCD) is widely used as a fractional factorial design in RSM optimization. It employs a simultaneous analysis of the individual and interaction effects of all parameters. To optimize the process parameters for the in vitro synthesis of soluble and active human TNF-α protein (the response), CCD was used.

#### 3.6.1. CCD Modelling

Cell-free expression of human TNF-α protein from the pET101/D-TOPO-human TNF-α recombinant plasmid was carried out using the RTS 100 *E. coli* HY kit according to the instructions in the manual. Therefore, the independent variables investigated in this CCD optimization experiment were limited to incubation temperature (A) and incubation time (B); each was examined at five different levels as shown in Table 4. The CCD matrix, in coded terms along with the cell-free synthesized human TNF-α concentration (estimated using quantitative ELISA) as the response is shown in Table 5.

Based on the experimental results, the following regression equation was obtained in coded form:*Y* = −3242.42 + 166.48 *A* + 286.61 *B* − 1.77 *A*^2^ − 0.38 *B*^2^ −9.14 *AB*(2)
where *Y* is the response (human TNF-α concentration in µg/mL), *A* and *B* are the coded terms for the independent variables. The positive and negative signs represent the synergistic and antagonistic effects on the response.

Different combinations of the independent variables yielded cell-free synthesized human TNF-α product ranging from 7.25 to 350.03 µg/mL as depicted in Table 5. The lowest value (7.25 µg/mL) of human TNF-α expression was obtained under the condition of incubation time (6 h) and incubation temperature (20 °C). While the highest value (350.03 µg/mL) was recorded at a certain incubation time (4 h) and incubation temperature (35 °C). 

Table 6 lists the regression coefficients and *p*-values calculated by the model for the linear, quadratic and interactions. The significance of each term of the model was verified using the associated *p*-value (*p* < 0.05). All the terms were statistically significant except for the quadratic term of incubation time (B × B) which was not significant. The *R*^2^, Adj *R*^2^ of the model were 86.6% and 77.1%, respectively.

#### 3.6.2. ANOVA

The results of ANOVA for the quadratic regression model obtained from CCD employed in the optimization of human TNF-α cell-free synthesis are summarized in Table 7. The *p*-value for the response surface quadratic model revealed that the regression is statistically significant. Moreover, all the model terms (linear, quadratic and interaction of independent variables) were all significant. Additionally, the *p*-value for the lack of fit of the model was not significant (*p* > 0.05). 

#### 3.6.3. Model Validation Using Residuals

The normal (%) probability plot of the residuals for the model is illustrated in Figure 9. The results revealed that the errors are normally distributed. The residuals (*ri*) are well distributed as positive and negative residuals in the range of −2 < *ri* < +2.

#### 3.6.4. Interactive Effect of Independent Variables

The interactive effect of the independent variables on the human TNF-α cell-free expression was obtained in the form of a 3D-surface plot (Figure 10A) and a 2D-contour plot (Figure 10B). The figures indicate that cell-free expression of human TNF-α is sensitive to incubation temperature as well as time. The 3D-surface plot demonstrates that, within the examined range, increasing the temperature affects the response positively. However, it is likely that long incubation times have adverse effects on the product. Moreover, the 2D-contour plot displays an elliptical nature, implying that the effect of the interaction on the human TNF-α expression is significant.

#### 3.6.5. Experimental Model Validation

With the purpose of confirming the optimized conditions, the cell-free expression experiment under optimum conditions was performed to validate the accuracy of the model. The predicted optimal values of the independent variables for human TNF-α cell-free expression are depicted in Table 8. The optimal condition has an incubation time of 2 h at a temperature of 40 °C. Approximately, a two-fold increase in human TNF-α cell-free production was achieved under the optimized condition (390 µg/mL) as compared to the basal (200 µg/mL) (Table 8).

### 3.7. Bioassays of Cell-Free Synthesized Human TNF-α Protein 

#### 3.7.1. Determination of the Cytotoxicity of Cell-Free Synthesized Human TNF-α against Normal Human Cells

The antiproliferative activity of cell-free synthesized human TNF-α on PBMCs was evaluated using the colorimetric MTT assay. As shown in Table 9 and Figure 11, treatment with human TNF-α caused a reduction in PBMCs viability in a dose-dependent manner. After 72 h of human TNF-α treatment, the IC_50_ and EC_100_ values, at which 50 and 100% of cells are metabolically active, were estimated as 1728.2 ± 10.74 and 250 ± 0.62 ng/mL, respectively (Table 10).

#### 3.7.2. Determination of the Cytotoxicity of Cell-Free Synthesized Human TNF-α against Human Cancer Cell Lines

MTT assay was carried out to assess the anticancer activity of cell-free synthesized human TNF-α against human cancer cell lines including Caco-2, HepG-2 and MCF-7. The expressed human TNF-α demonstrated cytotoxic effects in a concentration-dependent manner on Caco-2, HepG-2 and MCF-7 cells as shown in Table 11 and Figure 12, with IC_50_ values of 297.6 ± 10, 197.45 ± 0.05 and 373.93 ± 3.5 ng/mL, respectively (Table 12). It was observed that IC_50_ values of cell-free synthesized human TNF-α against test cancer cell lines were significantly (*p* < 0.05) lower than that against PBMCs. Additionally, the IC_50_ value of cell-free synthesized human TNF-α against HepG-2 was significantly (*p* < 0.05) lower than its EC_100_ value against PBMCs (Table 10 and Table 12). Thus, the cell-free synthesized human TNF-α showed significant anticancer potency and selective cytotoxicity against test cancer cell lines.

#### 3.7.3. Assessment of Cancer Cell Death Using EtBr-AO Nuclear Staining

The antitumor effects of human TNF-α protein on Caco-2, HepG-2 and MCF-7 cancer cell lines were investigated concerning the morphological shape of cells by fluorescence microscopy using an EtBr and AO dye mix. The AO is taken up by living as well as dead cells emitting green fluorescence if interrelated into the DNA, whereas EtBr is acquired only by the cells that have lost the integrity of their membrane and emit red fluorescence. According to the fluorescence emission (Figure 13B), early necroptotic cells with bright-green nuclei and late necroptotic cells with fragmented, orange to red nuclei were distinguished upon treatment with the IC_50_ of cell-free synthesized human TNF-α protein. Nevertheless, the untreated viable cells have uniform green nuclei with an organized structure (Figure 13A). This result indicated that cell-free synthesized human TNF-α protein induced cancer cell death.

## 4. Discussion

TNF-α is one of the key pro-inflammatory cytokines with a unique capability of inducing apoptotic cell death. The modulation of cell growth and death is a vital approach in treatment of cancer diseases. Thus, TNF-α has been used for malignant disease treatment since beginning of the twentieth century. Ever since the discovery of TNF-α, it has been comprehensively studied for its involvement in acute and chronic inflammation, cell differentiation and proliferation, stimulation of programs of cellular suicide including apoptosis and necroptosis, cell metabolism, autoimmune diseases and cancer [1,2,3,4,5].

The market demand for recombinant proteins manufacturing is increasing steadily, with a sale valued at USD 1.654 billion in 2017 and anticipated to rise to USD 2.850 billion by 2022 [42]. About half of this market is directed to production of recombinant pharmaceutical proteins (e.g., growth factors, antibodies, cytokines, vaccines and enzymes), followed by industrial proteins and research reagents [42,43]. Currently, most of the recombinant proteins are produced in prokaryotic cells (predominantly *E. coli*). However, other platforms are available but are less common such as mammalian cell lines, insect cells, algae, yeast and cell-free expression systems [25,43,44]. Additionally, some expression systems are still not significantly involved in commercial production of recombinant proteins such as systems of plants and their cells [43].

The use of cell-free expression systems is increasing in importance as their productivity increases and cost decreases. This approach provides unique opportunities not only for protein synthesis, but also for development of therapeutics, metabolic engineering, as well as research and education. In contrast to in vivo protein synthesis systems, CFPS platforms are of an open nature because expression occurs by use of cellular protein machinery outside living host cells without any constraints of cell membrane or other life objectives of the cell. This open nature allows for direct manipulation of cellular machinery for production of target and novel proteins, posttranslational modifications, or fluorescence labeling of proteins. Most importantly, CFPS avoids any host–target protein interactions, and therefore can produce proteins which are difficult to express in vivo because of their burden on the living cell, toxicity, or sensitivity to degradation by cellular enzymes. Such proteins comprise proteins from templates of high GC content, large proteins, antibodies and metalloproteins. Other applications of CFPS platforms include high-throughput screening, production of virus-like particles, membrane protein expression and continuous formats employed for large-scale reactions in industrial applications [45,46].

The aim of the present study was to construct and optimize an *E. coli* S30 extract-based cell-free system for the expression of soluble and active human tumor necrosis factor-alpha. To the best of our knowledge, this is the first report to produce rhTNF-α in CFPS system and optimize its expression by RSM. We previously carried out a similar application for the CFPS system based on *E. coli* lysate in the expression of another valuable cytokine: human consensus interferon-alpha [37].

We first evaded expression problems due to codon bias (incompatibility of codon usage between human sTNF-α gene and *E. coli* CFPS system) via computational optimization of the codon profile of the human sTNF-α gene as previously described for human sTNF-α and other recombinant proteins [7,22,36,47,48]. 

The codon-optimized synthetic human TNF-α gene was constructed by a two-step (assembly and amplification steps) PCR using codon-optimized and overlapped oligonucleotides which represent most of the sequence of both strands of the DNA duplex. In other studies, Binepal et al., 2012 [49] and Yin et al., 2017 [50] synthesized the human TNF-α gene using an overlap forward-primer-walk polymerase chain reaction (OFPW-PCR) and overlap extension PCR, respectively. Moreover, Castineiras et al., 2018 [22] reported the synthesis of the human TNF-α gene for recombinant expression of TNF-α protein.

Cell-free expression of human TNF-α protein from pET101/D-TOPO-human TNF-α recombinant plasmid was carried out using the RTS 100 *E. coli* HY kit. We chose *E. coli* crude cell extract as the CFPS system because of many factors including the absence of any posttranslational modification that is required for production of functionally active human TNF-α protein; reasonable productivity (2–20 µg/50 µL reaction) and cost (1.2 USD–12 USD/µg protein) of the system; cell-free reaction preparation being quick, simple and easy to optimize; high translation efficiency of the system and finally obtaining 390 µg/mL of soluble and functionally active human TNF-α protein in only 2 h at a temperature of 40 °C. However, this system suffers from some drawbacks such as an undefined system composition with presence of nucleases, proteases and tmRNA, low engineering flexibility and low tolerance for linear templates [34].

Although recombinant human TNF-α was previously produced in *E. coli* cells with a high yield of milligram (mg) protein per liter of bacterial culture [14,22], the expressed protein accumulated in inclusion bodies, which needed harsh denaturation processing, purification and protein refolding to obtain functionally active protein [51,52]. Lately, some reports claimed high-yield (7.2 mg/L to about 1.26 g/L of bacterial culture) production of soluble human TNF-α protein in the *E. coli* expression system after optimization of conditions of culture and induction [13,14]. The *E. coli* crude cell extract-based CFPS system used in the present study, on the contrary, produced about 390 µg/mL of soluble and functionally active human TNF-α protein in only 2 h at a temperature of 40 °C without need for cell lysis, protein extraction, refolding or other unnecessary processes. Higher yields at incubation time of hours and higher percent recovery can even be achieved if continuous formats of this system are used and optimized. Nevertheless, the yield obtained (70.5 µg/mL after purification) in the current work is a good starting point for scaling-up and optimizing the cell-free production of human TNF-α for future industrial or commercial purposes.

Experimental model validation of RSM revealed that the yield of cell-free production of rhTNF-α was nearly doubled under the optimized condition as compared to the basal condition. Experimental investigations implemented to examine the biological activity of the cell-free synthesized human TNF-α protein confirmed its significant anticancer potency and selective cytotoxicity against test cancer cell lines. Its selective cytotoxicity was observed from the fact that its IC_50_ values against test cancer cell lines were much lower than that against normal human PBMCs. Moreover, early necroptotic cells with bright-green nuclei and late necroptotic cells with fragmented, orange to red nuclei were distinguished upon treatment of test cancer cell lines with the IC_50_ of human TNF-α protein.

Human TNF-α protein has a single intramolecular disulfide bond between two cysteine residues at residues 69 and 101, which was successfully formed many times in *E. coli* in vivo expression systems without any problem or need for any cofactor [13,14,22]. *E. coli* and eukaryotic cells share the reducing cytoplasm that strongly disfavors the formation of stable disulfide bonds in proteins, but *E. coli* cells do not have structures resembling the eukaryotic endoplasmic reticulum. Instead of this, *E. coli* has an oxidizing periplasm. Therefore, eukaryotic protein expression in bacterial periplasm (especially in commercially available engineered competent bacterial strains) is possible. As this cytokine has only a single intramolecular disulfide bond, it is successfully formed in bacteria, not only in *E. coli* but also in *Streptomyces lividans* [16]. The problem appears and the need for optimizing strategies with expression of recombinant eukaryotic proteins with multiple disulfide bonds in *E. coli* as reported by de Marco, 2009 [53] and Ma et al., 2020 [54]. The used expression system in the present study is *E. coli* chemical protein synthesis machinery not cells, which is the innovation; however, the capabilities of living cells are still available, combined with avoiding the negatives of cell-confining membranes and the possibility of having reducing agent(s) to maintain reduced conditions. We used *E. coli* and its cell-free system before in production of another recombinant cytokine with a single disulfide bond, human consensus interferon-alpha, and obtained functional antiviral and anticancer protein [36,37,55]. Although cell-free cytokine production is a successful approach, *E. coli* cell-free systems and their in vivo counterparts share the same problem, which is a limited ability to correctly fold proteins containing multiple disulfide bonds, which is not the case here [56,57]. Moreover, the bioactivity of tumor necrosis factor is dependent on the disulfide bond as reported by Narachi et al., 1987 [58]; thus, the anticancer potency of cell-free synthesized human TNF-α protein in cell assays described in this study confirmed successful formation of the bond.

## 5. Conclusions

Overall, we suggest the use of an *E. coli* crude cell extract-based CFPS system for rapid expression of soluble and functionally active human TNF-α protein, which can provide the requirements for research and therapeutic applications of TNF-α.

## Figures and Tables

**Figure 1 biology-11-00157-f001:**
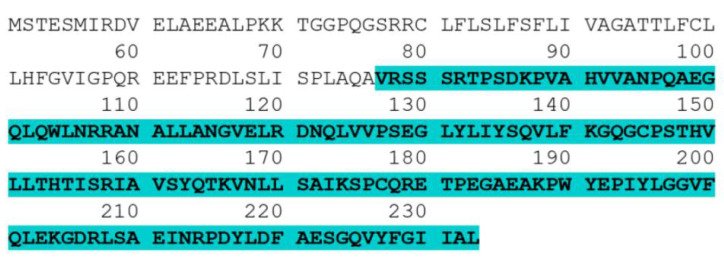
Complete amino acid sequence of human TNF-α. The highlighted part (amino acids 77–233) is the sequence of the human TNF-α soluble form.

**Figure 2 biology-11-00157-f002:**
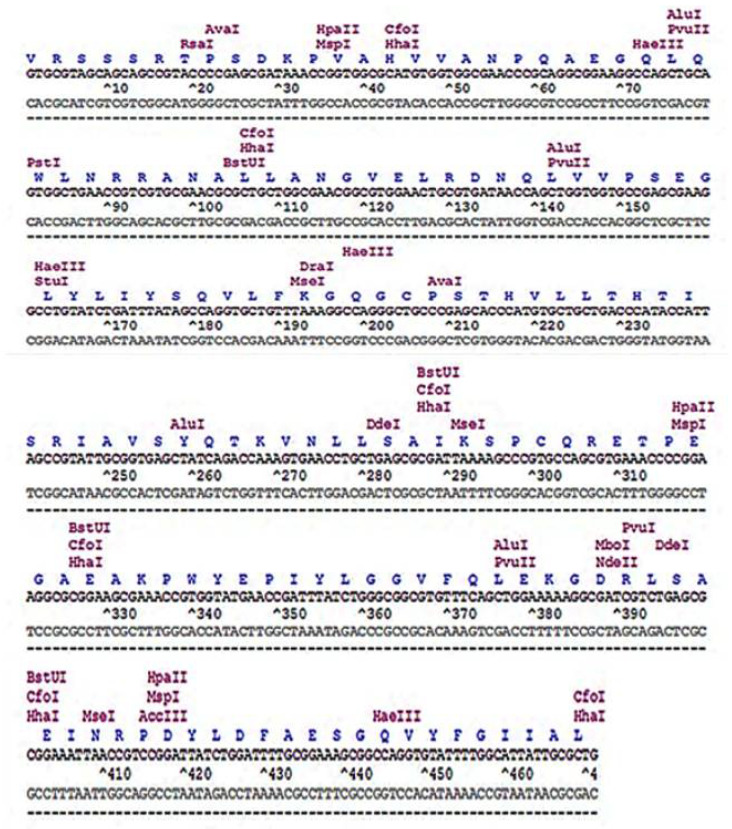
Restriction map of the codon-optimized human TNF-α coding sequence.

**Figure 3 biology-11-00157-f003:**
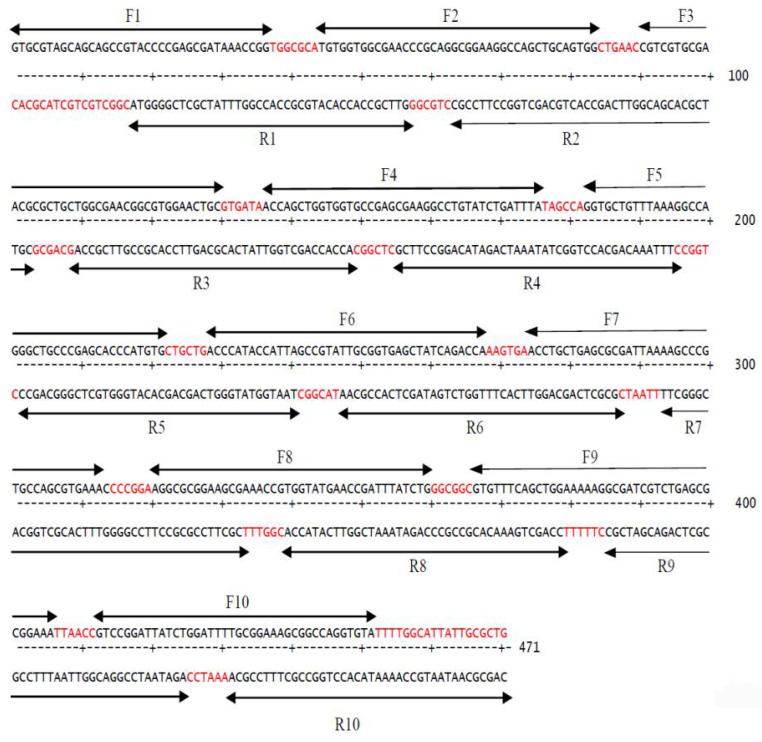
Positions of oligonucleotides used in human TNF-α synthetic gene assembly indicated by arrows. Gaps in between the designed oligonucleotides are in red color.

**Figure 4 biology-11-00157-f004:**
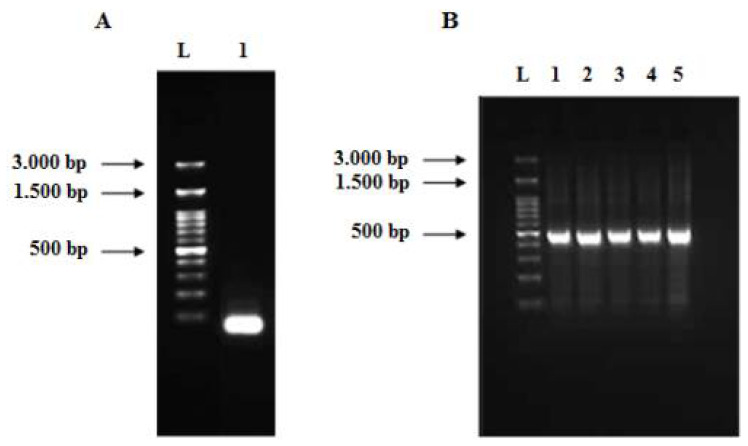
Agarose gels (2%) of the assembled PCR product (**A**) and full-length human TNF-α gene (**B**). (**A**) Lane L: 100 bp DNA ladder and lane 1: assembled PCR product of the human TNF-α gene. (**B**) Lane 1 to 5: full-length human TNF-α gene (474 bp) at different annealing temperatures (57, 57.5, 58, 58.5, 59 °C), respectively.

**Figure 5 biology-11-00157-f005:**
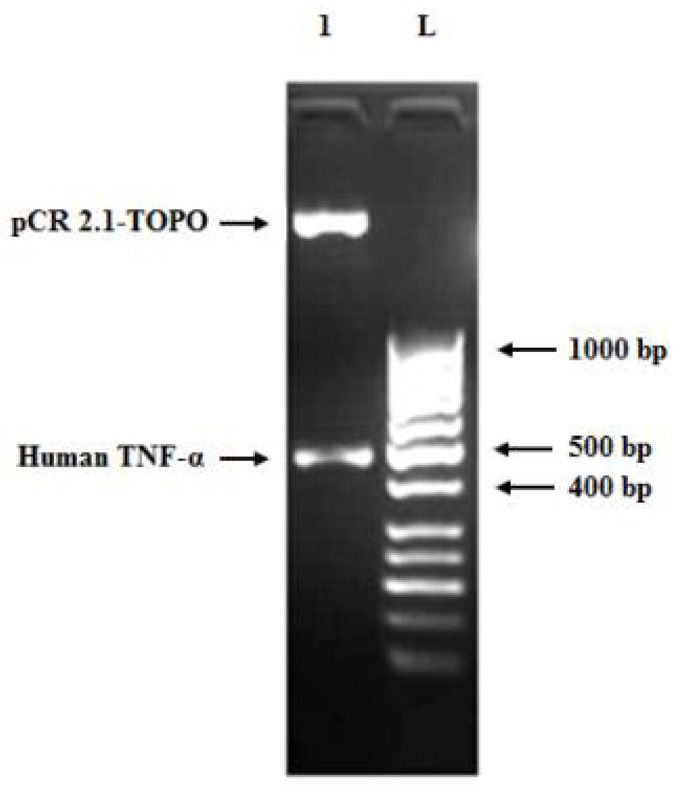
Agarose gel (1.2%) of restriction digestion of a positive clone carrying pCR 2.1-TOPO-human TNF-α recombinant plasmid using *EcoR*I enzyme at 37 °C for 1 h. Lane L: 50 bp DNA ladder and lane 1: pCR 2.1-TOPO vector fragment (3.881 bp) and human TNF-α insert band (505 bp, comprising full-length gene size of 474 bp, restriction enzymes sites of 12 bp and 19 bp from the vector cloning site).

**Figure 6 biology-11-00157-f006:**
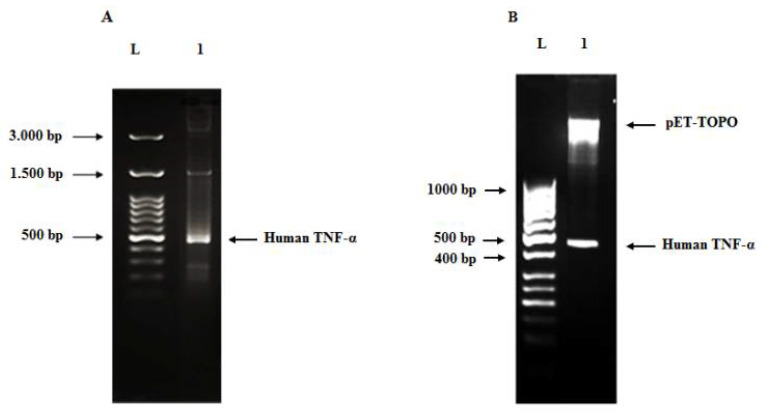
Agarose gel (2%) of human TNF-α PCR product from recombinant plasmid (**A**), agarose gel (1.2%) of restriction digestion of pET101/D-TOPO-human TNF-α plasmid using *BamH*I and *Xho*I enzymes at 37 °C for 1 h (**B**). Lane L: DNA ladder, lane 1 (**A**): human TNF-α amplicon (490 bp) amplified from the pET101/D-TOPO-human TNF-α recombinant plasmid and lane 1 (**B**): pET101/D-TOPO vector fragment (5.763 bp) and human TNF-α insert band (480 bp).

**Figure 7 biology-11-00157-f007:**
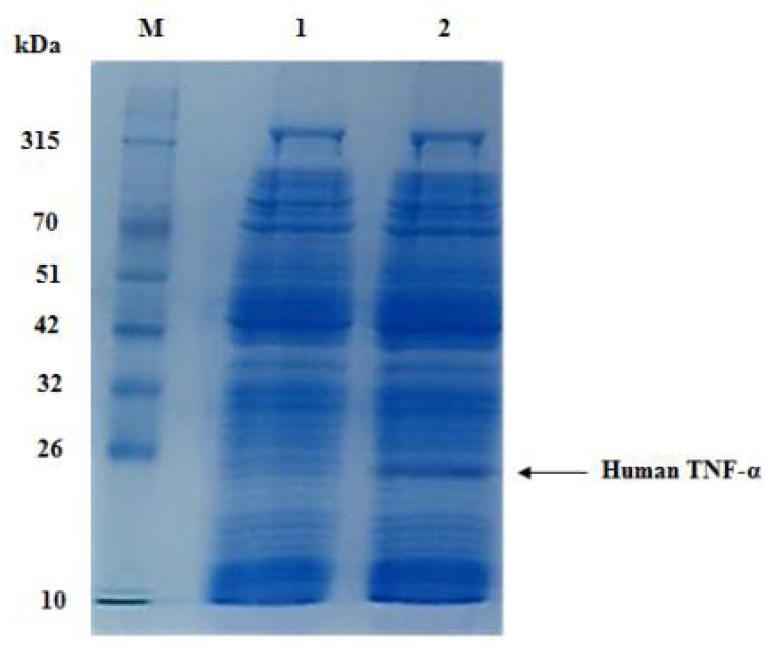
SDS-PAGE analysis of cell-free synthesized human TNF-α protein at 30 °C within 6 h. Lane M: a prestained protein marker (10–315 kDa), lane 1: negative control reaction and lane 2: cell-free synthesized human TNF-α protein (21 kDa).

**Figure 8 biology-11-00157-f008:**
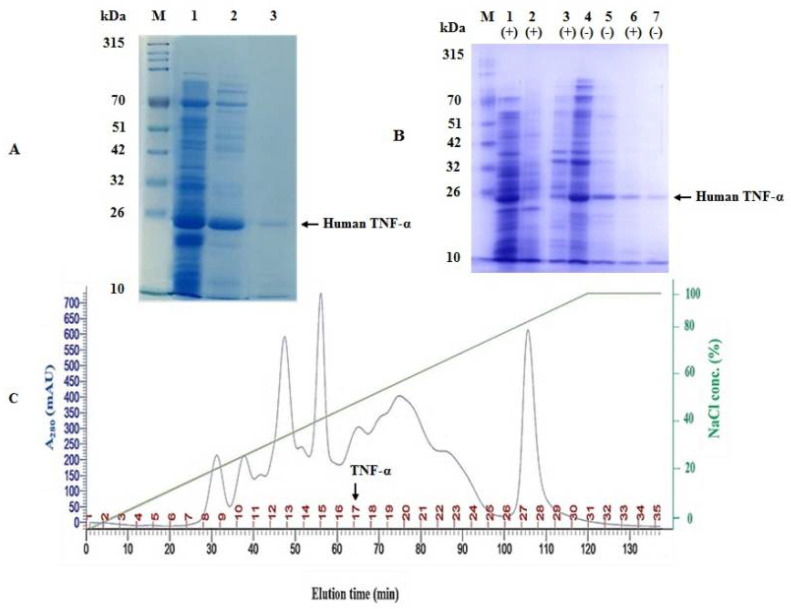
Two-step chromatographic purification of cell-free synthesized human TNF-α protein. (**A**) SDS-PAGE examination of the purification steps of cell-free synthesized human TNF-α protein. Lane M: a prestained protein marker (10–315 kDa), lane 1: the soluble protein fraction of CFPS reaction of human TNF-α, lane 2: sample from peak 17 in the elution profile and lane 3: pooled elution fraction of second purification step using heparin-Sepharose 6 fast flow affinity column. (**B**) SDS-PAGE examination of the purification steps of cell-free synthesized human TNF-α protein in reducing (+) and non-reducing (−) conditions. Lane M: a prestained protein marker (10–315 kDa), lanes 1 and 4: the soluble protein fraction of CFPS reaction of human TNF-α, lane 2: negative control reaction, lanes 3 and 5: sample from peak 17 in the elution profile and lanes 6 and 7: pooled elution fraction of second purification step using heparin-Sepharose 6 fast flow affinity column. (**C**) Chromatographic (elution) profile for purifying human TNF-α protein by ion exchange chromatography column packed with Q-Sepharose fast flow resin.

**Figure 9 biology-11-00157-f009:**
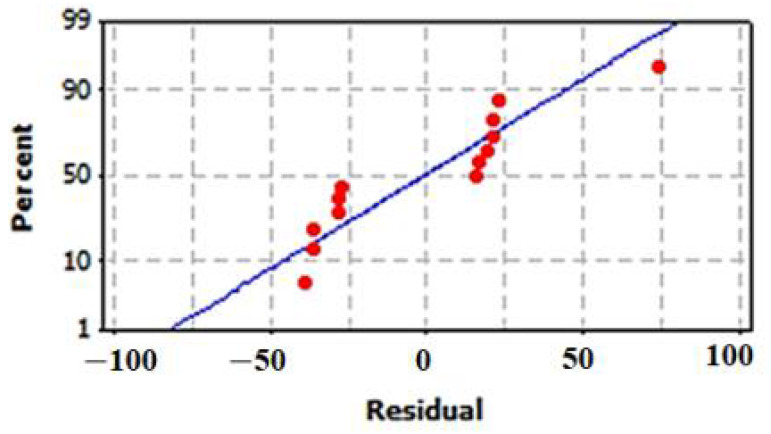
Normal probability (%) plot of the residuals for the model of cell-free synthesized human TNF-α.

**Figure 10 biology-11-00157-f010:**
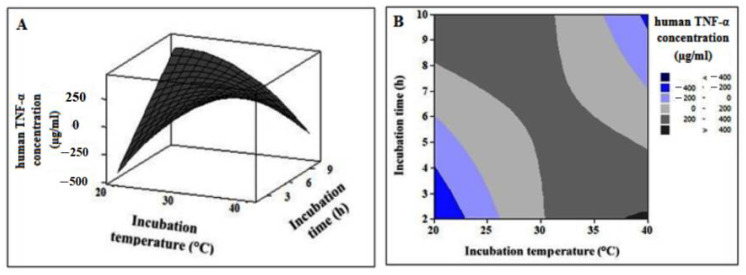
The 3D-surface plot (**A**) and 2D-contour plot (**B**) showing the interactive effect of independent variables on human TNF-α cell-free synthesis.

**Figure 11 biology-11-00157-f011:**
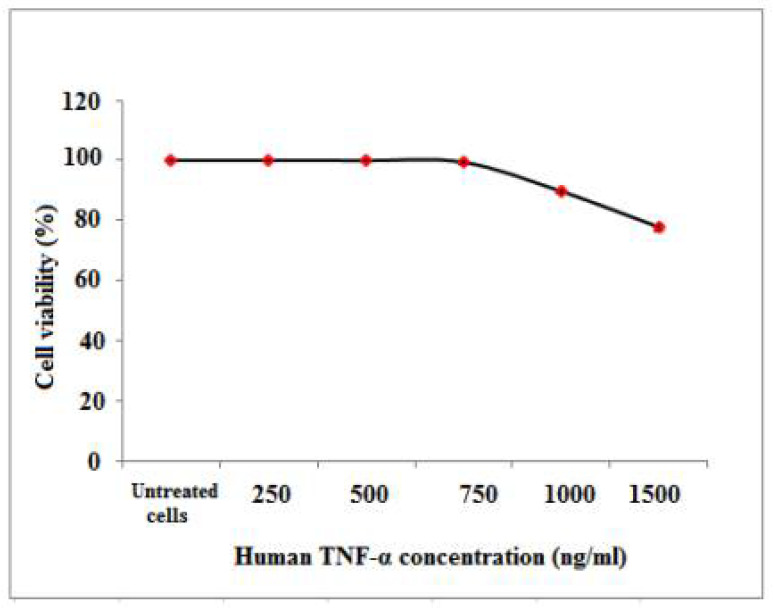
Percentage of cell viability of PBMCs upon treatment with different cell-free synthesized human TNF-α concentrations please remove the mouse, we uploaded original image.

**Figure 12 biology-11-00157-f012:**
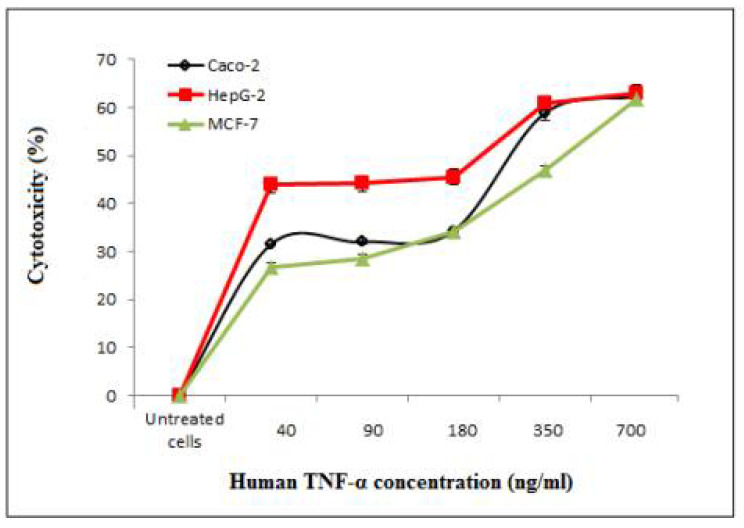
Inhibitory effect of cell-free synthesized human TNF-α on the proliferation of Caco-2, HepG-2 and MCF-7 human cancer cell lines.

**Figure 13 biology-11-00157-f013:**
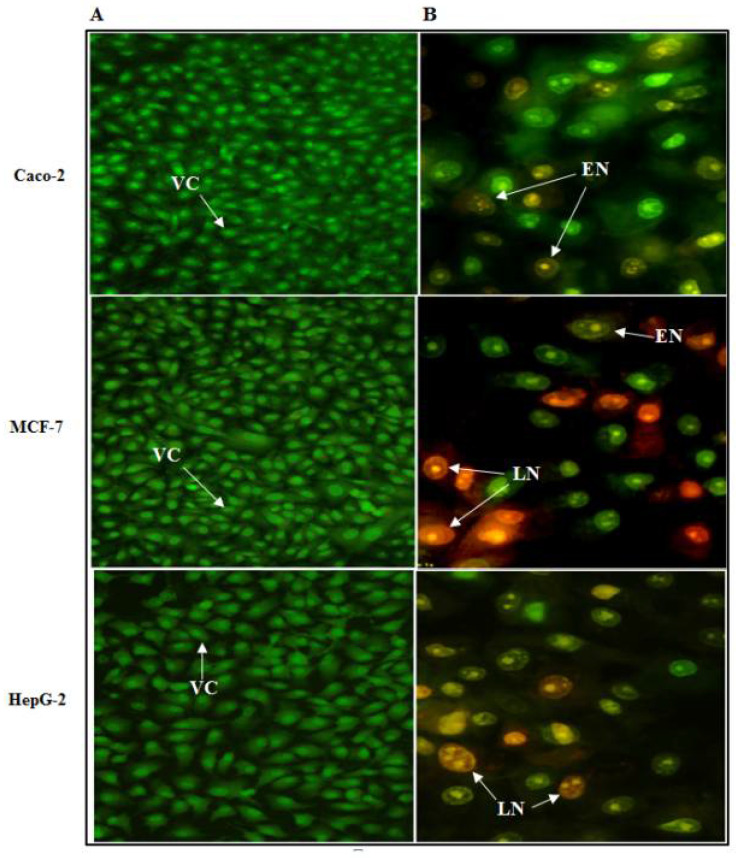
Fluorescence microscopy images of Caco-2, HepG-2 and MCF-7 cells stained with EtBr/AO dye mix. (**A**) Untreated control cells and (**B**) cells after 72 h treatment with the IC_50_ of cell-free synthesized human TNF-α protein. VC: Viable cells, EN: early necroptotic cells and LN: late necroptotic cells (magnification × 400).

**Table 1 biology-11-00157-t001:** Rare codon analysis results of DNA coding sequence for human TNF-α.

Rare Codon Analysis		
	Index	Parameter
Codon adaptation index (CAI)	0.7	0.8–1.0
Guanine-cytosine (GC) content	68.93%	30–70%
Codon frequency distribution (CFD)	Percentage of low frequency codons is 12%	<30%

**Table 2 biology-11-00157-t002:** Sequences of human TNF-α gene-specific primers.

Primer	Vector	Sequence
Forward	pCR 2.1-TOPO	5′-GGATCCATGGTGCGTAGCAGCA-3′
Forward	pET101/D-TOPO	5′-CACCGGATCCATGGTGCGTAGC-3′
Reverse	pCR 2.1-TOPO and pET101/D-TOPO	5′-CTCGAGCAGCGCAATAATGCC-3′

The underlined sequences indicate *BamH*I and *Xho*I restriction sites.

**Table 3 biology-11-00157-t003:** Purification of cell-free synthesized human TNF-α protein by the two-step chromatographic strategy.

Purification Step	Total Protein Concentration (µg/mL)	Recovery (%)
Soluble fraction of CFPS reaction	320.7	100
Q-Sepharose	160.3	49.98
Heparin–Sepharose	70.5	21.98

**Table 4 biology-11-00157-t004:** Coded values of variables used in CCD for the optimization of cell-free synthesis of human TNF-α.

Independent Variables	Levels				
	−−	−	0	+	++
A: Incubation temperature ( °C)	20	25	30	35	40
B: Incubation time (h)	2	4	6	8	10

**Table 5 biology-11-00157-t005:** Central composite matrix with experimental and predicted results of cell-free synthesized human TNF-α.

Run Order	Independent Variables		Response Human TNF-α Concentration (µg/mL)		Standardized Residual
	A	B	Actual	Predicted	
1	0	0	191.7	218.8	−0.66
2	++	0	57.0	95.5	−1.86
3	0	0	242.2	218.8	0.57
4	−	+	217.0	253.1	−1.12
5	0	++	260.9	244.0	0.81
6	0	0	191.0	218.8	−0.67
7	0	0	240.0	218.8	0.52
8	−	−	55.6	39.0	0.52
9	−−	0	7.2	−12.1	0.93
10	0	−−	145.3	181.3	−1.74
11	0	0	190.7	218.8	−0.68
12	+	+	145.9	124.2	0.67
13	+	−	350.0	275.6	2.31

**Table 6 biology-11-00157-t006:** Estimated regression coefficients of human TNF-α cell-free synthesis.

Term	Coefficient	Standard Error of Coefficient (SE Coef)	*T*	*P*
Constant	−3242.42	551.14	−5.88	0.001
A	166.48	26.56	6.27	0
B	286.61	73.84	3.88	0.006
A × A	−1.77	0.38	−4.68	0.002
B × B	−0.38	2.36	−0.16	0.876
A × B	−9.14	2.26	−4.04	0.005

**Table 7 biology-11-00157-t007:** ANOVA results for the response surface quadratic model of the human TNF-α cell-free synthesis.

Source	Degree of Freedom (DF)	Sequential Sum of Squares (Seq SS)	Adjusted Sum of Squares (Adj SS)	Adjusted Mean Squares (Adj MS)	*F*	*P*
Regression	5	93,002	93,002	18,600.4	9.09	0.006
Linear	2	11,638	80,774	40,387.2	19.73	0.001
Square	2	47,956	47,956	23,978.0	11.71	0.006
Interaction	1	33,409	33,409	33,408.5	16.32	0.005
Residual error	7	14,329	14,329	2047.0		
Lack of fit	3	11,326	11,326	3775.4	5.03	0.076
Pure error	4	3003	3003	750.7		
Total	12	107,331				

**Table 8 biology-11-00157-t008:** Recombinant human TNF-α cell-free synthesis before and after optimization.

Independent Variables	Levels		Human TNF-α Concentration (µg/mL) Estimated by Quantitative ELISA	
	Basal	Optimum	Basal	Optimum
Incubation temperature (°C)	30	40	200	390
Incubation time (h)	6	2		

**Table 9 biology-11-00157-t009:** Effect of different cell-free synthesized human TNF-α concentrations on viability of PBMCs.

Human TNF-α Concentration (ng/mL)	Percentage Cell Viability
250	99.99 ± 0.01
500	99.96 ± 0.003
750	99.37 ± 0.56
1000	89.53 ± 0.48
1500	77.45 ± 1.09

Experiment was performed three times, each in duplicates. Data are represented as the mean ± standard error of the mean (SEM).

**Table 10 biology-11-00157-t010:** IC_50_ and EC_100_ of cell-free synthesized human TNF-α on PBMCs.

Human TNF-α Concentration (ng/mL)	
IC_50_	EC_100_
1728.24 ± 10.74	250.08 ± 0.62

Results are represented as means ± SEM.

**Table 11 biology-11-00157-t011:** Cytotoxicity of cell-free synthesized human TNF-α on human cancer cell lines.

Human TNF-α Concentration (ng/mL)	Percentage Cytotoxicity		
	Caco-2	HepG-2	MCF-7
40	31.50 ± 0.10	43.88 ± 0.14	26.82 ± 0.91
90	32.01 ± 0.60	44.17 ± 0.39	28.53 ± 0.94
180	34.28 ± 0.59	45.58 ± 0.29	34.20 ± 1.27
350	58.81 ± 1.56	60.86 ± 0.30	46.80 ± 0.80
700	62.33 ± 0.15	63.11 ± 0.97	61.82 ± 0.68

The experiment was performed three times, each time in duplicate. Data are represented as means ± SEM.

**Table 12 biology-11-00157-t012:** IC_50_ and EC_100_ of cell-free synthesized human TNF-α against human cancer cell lines.

Human TNF-α Concentration (ng/mL)	Human Cancer Cell Lines		
	Caco-2	HepG-2	MCF-7
IC_50_	297.6 ± 10	197.4 ± 0.05	373.9 ± 3.5
EC_100_	17.71 ± 0.07	10 ± 0.02	10 ± 1.63

Data are represented as means ± SEM.

## Data Availability

Data are contained within the article.

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
