# Peer review of "A De Novo Optimized Cell-Free System for the Expression of Soluble and Active Human Tumor Necrosis Factor-Alpha"

_biology, 2022, doi:10.3390/biology11020157_

Round 1

Reviewer 1 Report

The paper is too long, and much information must be reorganized in the supporting material, which is a missing part. The expression of the cytokine, which should require disulfide bond formation is not explained clearly in the paper and is a major concern in the manuscript. Some data about the gene synthesis must be reported well and cannot be published as a current result. The cell assays are not clearly described. In conclusion, your work appears too misleading to me, and I don’t support the publication of this manuscript. If interested, below you can find the comments that I wrote going over your paper.

You have mentioned that eukaryotic systems are better suited for sTNF-alpha expression due to the possible PTMs with such systems. However, you haven’ t indicate which PTM are needed for sTNF-alpha or if any are needed. Please clarify this point. It seems that in the soluble protein there is a disulfide bond.

Introduce earlier figure 3, it is necessary to understand better the strategy you used to synthesize the gene.

It is not clear while reading the paper which gene you are building into the plasmid for gene expression. You mention at the beginning that you are going to express sTNF-alpha and then you produce the plasmid pET-hTNF-alpha, and later you write that you are going to express rhTNF-alpha using the E. coli cell-free expression system. Clarify this point. 

The kit you are using is not designed for the expression of proteins with disulfide bonds. There’s an important amount of literature that explain that the expression of cytokine using E. coli cell-free expression systems requires the addition of co-factors to the reaction such as chaperone and glutathione buffer. How did you express the sTNF-alpha without co-factors in the reaction?   

Why did you perform the acetone precipitation? How did you obtain the soluble fraction of the protein of interest?

Figure 1 and table 1 should go in the supporting material. Figure 1 is a poor-quality image, please upload an image with better resolution.

Please move in the supporting material Figure 2, Figure 3 and table 2. Upload better quality images, especially for Figure 2.  

Omit in the figure legend the following sentence: “which were not modified in 375 any way”. It’s not conceivable that data are modified.

Figure 4 A doesn’t make sense. You mention that it is the assembled product from the oligos assembly but the band you show in the gel is too short to be the assembly product. When describing the result of Figure 4A specify which is the expected length of the synthesized product.

Figure 4B how do you see that the annealing temperature at 58C produced the best result? This cannot be drawn from the data shown in the figure.

You must show the result of DNA sequencing of the positive clone carrying your gene of interest. This must be shown in the SI.

Specify if the ELISA assay used to quantify the protein (200 µg/mL) was performed with the protein still in the lysate before purification. However, after CFPS you should centrifuge the lysate and collect the supernatant for ELISA quantification. Why didn’t you perform this step?

Figure 8B should go in the supporting material.

Table 3 the soluble fraction is 320.68 µg/mL? You have mentioned that after CPFS you quantified 200 µg/mL through ELISA assay. Clarify this point?

Move table 6 in the supporting information. What is the conclusion of this analysis, i.e., the quadratic term of incubation time was not significant? Therefore, the incubation time which relevance has on the result of your experiment?

Table 7 in the supporting material.

Figure 9 is of poor quality and the analysis shown is irrelevant for such a simple optimization problem. Remove it from the paper. Report in word that your model has been validated by using residuals, in particular, which kind of residual.

Table 9 in supporting material.

Explain briefly what a colourimetric MTT assay is. At least the acronym.

How did you estimate the IC50 and EC100? Show data plot in the paper.  

Figure 11 is blurry upload better quality plot.

Table 11 in supporting – data are represented in the plot in figure 12.

Show all the data to calculate the IC50 and E100

The assay depicted in Fig. 13 it’s not convincing, the difference between the to set of A and B of cell samples

Cite this paper in the discussion: Microscale to Manufacturing Scale-up of Cell-Free Cytokine Production—A New Approach for Shortening Protein Production Development Timeline

Biotechnol Bioeng. 2011 Jul; 108(7): 1570–1578.

Also, explain which cofactors are needed for disulfide bond formation. Especially for cytokines, this is not mentioned in the whole paper.

Author Response

Dear Editor in chief

Dear Editor

Dear Reviewers

I would like to express our deep gratitude for all positive comments and suggestions made by the respected reviewers and our respect for negative ones. I will defend our work as possible in response to the negative comments and concerns of Reviewer 1.

Here is the simple summary:

Simple Summary

As a result of increasing demand for the pleiotropic cytokine TNF-α, recombinant human TNF-α protein with appropriate bioactivities was produced in several heterologous in vivo expression systems. While in vivo expression of this cytokine is laborious and lengthy, cell-free or in vitro expression system has the benefits of speed, simplicity, flexibility, focus of all the system energy on target protein synthesis alone, besides high soluble and functional protein yield. Therefore, we employed and optimized E. coli-based cell-free system for the first time to express recombinant human TNF-α. Our findings revealed that cell-free expression system can be an alternative platform for producing soluble and functionally active recombinant TNF-α with a yield of 390 µg/ml in only 2 h at a temperature of 40°C for further research and clinical trials.

Kindly, find below the response to all comments made by the three reviewers point by point.

Response to reviewer comments:

Reviewer 1:

Dear Reviewer:

Thank you for reviewing our manuscript. I will try to ease your concerns in the following responses to your comments.

Comment 1: The paper is too long, and much information must be reorganized in the supporting material, which is a missing part. The expression of the cytokine, which should require disulfide bond formation is not explained clearly in the paper and is a major concern in the manuscript.

Response: regarding the major concern which is disulfide bond formation, I will defend our work in six major points:  

[1] Human tumor necrosis factor has single intramolecular disulfide bond between 2 cysteine residues at residues 69 and 101, that was successfully formed many times in E. coli in vivo systems without any problem or need for any cofactor (most recent reports are cited as references [13], [14], [22] in the manuscript). We even have unpublished data for expression of the cytokine in E. coli living cells using three different induction strategies with functional protein yield.   

[2] E. coli shares with eukaryotic cells the reducing cytoplasm that strongly disfavors the formation of stable disulfide bonds in proteins, but do not have structures resembling eukaryotic endoplasmic reticulum. Instead of it, E. coli has an oxidizing periplasm. Therefore, eukaryotic protein expression in bacterial periplasm (especially in engineered competent strains commercially available) is possible. As this cytokine has only a single intramolecular disulfide bond, it is successfully formed in bacteria not only E. coli but also in Streptomyces lividans (reference [16]). The problem appears and the need for cofactors with Multiple Disulfide Bonds as reported in

de Marco A. Strategies for successful recombinant expression of disulfide bond-dependent proteins in Escherichia coli. Microb Cell Fact. 2009;8:26. Published 2009 May 14. doi:10.1186/1475-2859-8-26

Ma Y, Lee CJ, Park JS. Strategies for Optimizing the Production of Proteins and Peptides with Multiple Disulfide Bonds. Antibiotics (Basel). 2020 Aug 26;9(9):541. doi: 10.3390/antibiotics9090541.

[3] The used system in our study is E. coli chemical protein synthesis machinery not cells, which is the innovation but still the capabilities of living cells are available combined with avoiding negatives of their confining membranes and reductases.

[4] We used E. coli and its cell-free system before in production of another cytokine with a single disulfide bond, human consensus interferon alpha and obtained functional antiviral, anticancer cytokine (references [36], [37]). Note that activity of interferon is dependent on the bond.

Mohammed, Y.; EL-Baky, N.A.; Redwan, E.M. Expression, purification, and characterization of recombinant human consensus interferon-alpha in Escherichia coli under λPL promoter. Preparative Biochemistry and Biotechnology 2012, 42(5), 426-447, DOI: 10.1080/10826068.2011.637600.

El-Baky, N.A.; Omar, S.H.; Redwan, E.M. The anti-cancer activity of human consensus interferon-alpha synthesized in cell-free system. Protein Expression and Purification 2011, 80, 61–67, doi:10.1016/j.pep.2011.07.003

[5] Again, E. coli cell-free systems share the same problem with their in vivo counterparts (a limited ability to correctly fold proteins containing multiple disulfide bonds, which is not the case here) as discussed in the following study.

Despite many promising aspects of cell-free systems, several obstacles have previously limited their use as a protein production technology. These obstacles have included short reaction durations of active protein synthesis, low protein production rates, and difficulty in supplying the intense energy and substrate needs of protein synthesis without deleterious concomitant changes in the chemical environment. Furthermore, expensive reagent costs (particularly high energy phosphate chemicals in the form of nucleotides and secondary energy sources), small reaction scales, a limited ability to correctly fold proteins containing multiple disulfide bonds, and its initial development as a “black-box” science
were limitations (Swartz, 2006).

Swartz J. Developing cell-free biology for industrial applications. J Ind Microbiol Biotechnol. 2006 Jul;33(7):476-85. doi: 10.1007/s10295-006-0127-y. Epub 2006 May 9. PMID: 16761165.

[6] The activity of tumor necrosis factor is dependent on the bond (Narachi et al., 1987), thus the activity in cell assays reported in the manuscript as well as other still unpublished bioassays we performed, confirm successful formation of the bond.

Narachi MA, Davis JM, Hsu YR, Arakawa T. Role of single disulfide in recombinant human tumor necrosis factor-alpha. J Biol Chem. 1987 Sep 25;262(27):13107-10. PMID: 3654604.

Comment 2: You have mentioned that eukaryotic systems are better suited for sTNF-alpha expression due to the possible PTMs with such systems.

Response: The statement is general for TNF and its products. As I mentioned above and in the introduction it is non-glycosylated and with a single disulfide bond that can be formed in both prokaryotic and eukaryotic systems. However, the eukaryotic system is more efficient especially for fusion recombinant products of this cytokine  

Comment 3: Introduce earlier figure 3, it is necessary to understand better the strategy you used to synthesize the gene.

Response: we have previously demonstrated our strategy in details for artificial gene synthesis using PCR in [36] cited in methods section in Primers design for the synthesis of codon-optimized human TNF-α gene.

Comment 4: It is not clear while reading the paper which gene you are building into the plasmid for gene expression.

Response: we mentioned at the beginning sTNF-alpha is the target protein because there is another form of the cytokine (membranous), and we have artificially synthesized the soluble active form of it. Through the paper it is named as recombinant human TNF-alpha protein.

Comment 5: The kit you are using is not designed for the expression of proteins with disulfide bonds. There’s an important amount of literature that explain that the expression of cytokine using E. coli cell-free expression systems requires the addition of co-factors to the reaction such as chaperone and glutathione buffer. How did you express the sTNF-alpha without co-factors in the reaction?   

Response: Again, E. coli cell-free systems share the same problem with their in vivo counterparts (a limited ability to correctly fold proteins containing multiple disulfide bonds), which is not the case with this cytokine. Thus, no need for chaperone or glutathione buffer.

Comment 6: Why did you perform the acetone precipitation? How did you obtain the soluble fraction of the protein of interest?

Response: This is the manual instructions for soluble protein precipitation. Centrifuge reaction, separate supernatant, precipitate with acetone, then examine.  

Comment 7: Figure 1 and table 1 should go in the supporting material. Figure 1 is a poor-quality image, please upload an image with better resolution. Please move in the supporting material Figure 2, Figure 3 and table 2. Upload better quality images, especially for Figure 2. Figure 8B should go in the supporting material. Move table 6 in the supporting information. Table 7 in the supporting material. Figure 9 is of poor quality. Table 9 in supporting material. Figure 11 is blurry upload better quality plot.

Response: Unfortunately, we cannot remove data to supplementary material as this will significantly disrupt the design and flow of the article for the readers. The resolution of all images changed after reformatting of the paper by the journal.

Comment 8: Omit in the figure legend the following sentence: “which were not modified in any way”. It’s not conceivable that data are modified.

Response: we added this statement because we did not submit other forms of the images.

Comment 9: Figure 4 A doesn’t make sense. Figure 4B how do you see that the annealing temperature at 58C produced the best result? This cannot be drawn from the data shown in the figure.

Response: the assembled product is much shorter than amplified full-length, as it is an intermediate product. Regarding annealing temperature, the difference on the gel image is nonspecific products that were lower at this annealing temperature

Comment 10: Specify if the ELISA assay used to quantify the protein (200 µg/mL) was performed with the protein still in the lysate before purification.

Response:  we performed Centrifugation for CFPS reaction, separate supernatant, precipitate with acetone, then examine. In all cases, the kit is quantitative sandwich ELISA specific for the cytokine.   

Comment 11: Table 3 the soluble fraction is 320.68 µg/mL? You have mentioned that after CPFS you quantified 200 µg/mL through ELISA assay. Clarify this point?

Response: Table 3 the soluble fraction is 320.68 µg/mL: this is total protein content measured by Bradford assay from the CFPS reaction before purification. CPFS you quantified 200 µg/mL through ELISA assay: this is TNF content from the CFPS reaction before purification. Both detailed in methods. 

Associate Professor Dr Nawal Abd EL-Baky

Reviewer 2 Report

In the manuscript ‘De novo optimized cell-free system for the expression of soluble and active human tumor necrosis factor-alpha’, N.A. El-Baky et al reported in vitro expression of TNF-a by using cell-free expression system. The rhTNF-a showed good yield in CFES, and the purified protein showed good reactivity in cell assays.

Suggest to accept in current form.

Author Response

Dear Editor in chief

Dear Editor

Dear Reviewers

I would like to express our deep gratitude for all positive comments and suggestions made by the respected reviewers and our respect for negative ones. I will defend our work as possible in response to the negative comments and concerns of Reviewer 1.

Here is the simple summary:

Simple Summary

As a result of increasing demand for the pleiotropic cytokine TNF-α, recombinant human TNF-α protein with appropriate bioactivities was produced in several heterologous in vivo expression systems. While in vivo expression of this cytokine is laborious and lengthy, cell-free or in vitro expression system has the benefits of speed, simplicity, flexibility, focus of all the system energy on target protein synthesis alone, besides high soluble and functional protein yield. Therefore, we employed and optimized E. coli-based cell-free system for the first time to express recombinant human TNF-α. Our findings revealed that cell-free expression system can be an alternative platform for producing soluble and functionally active recombinant TNF-α with a yield of 390 µg/ml in only 2 h at a temperature of 40°C for further research and clinical trials.

Kindly, find below the response to all comments made by the three reviewers point by point.

Response to reviewer comments:

Reviewer 2:

Comment: In the manuscript ‘De novo optimized cell-free system for the expression of soluble and active human tumor necrosis factor-alpha’, N.A. El-Baky et al reported in vitro expression of TNF-a by using cell-free expression system. The rhTNF-a showed good yield in CFES, and the purified protein showed good reactivity in cell assays. Suggest to accept in current form.

Response: We are grateful for reviewing and suggesting acceptance of our manuscript for publication.

Associate Professor Dr Nawal Abd EL-Baky

Reviewer 3 Report

This paper will be improved by adding (1) the analysis of the disulfide bond that is formed in the active form of TNF-alpha and (2) activity comparison with a reference TNF-alpha sample. I think (1) is necessary while (2) is optional.

(1) They used a cell-free expression system from E. coli, which probably contains reducing agent(s) to maintain reduced conditions. Thus, the authors have to show if their TNF-alpha sample contains the disulfide bond to what extent. To do this, there are several possible methods: thiol quantification using DTNP/DTNB, comparison of reducing and non-reducing SDS-PAGE patterns, MALDI-TOF MS analysis of the trypsin digested fragments, reverse-phase HPLC analysis of S-alkylated proteins in non-reduced and reduced conditions.

(2) Recombinant TNF-alpha is commercially available. They should compare the activities of their sample and other product(s).

Figure 8 B  Indicate which peak is the peak of TNF-alpha using an arrow.

Author Response

Dear Editor in chief

Dear Editor

Dear Reviewers

I would like to express our deep gratitude for all positive comments and suggestions made by the respected reviewers and our respect for negative ones. I will defend our work as possible in response to the negative comments and concerns of Reviewer 1.

Here is the simple summary:

Simple Summary

As a result of increasing demand for the pleiotropic cytokine TNF-α, recombinant human TNF-α protein with appropriate bioactivities was produced in several heterologous in vivo expression systems. While in vivo expression of this cytokine is laborious and lengthy, cell-free or in vitro expression system has the benefits of speed, simplicity, flexibility, focus of all the system energy on target protein synthesis alone, besides high soluble and functional protein yield. Therefore, we employed and optimized E. coli-based cell-free system for the first time to express recombinant human TNF-α. Our findings revealed that cell-free expression system can be an alternative platform for producing soluble and functionally active recombinant TNF-α with a yield of 390 µg/ml in only 2 h at a temperature of 40°C for further research and clinical trials.

Kindly, find below the response to all comments made by the three reviewers point by point.

Response to reviewer comments:

Reviewer 3:

Comment 1: They used a cell-free expression system from E. coli, which probably contains reducing agent(s) to maintain reduced conditions. Thus, the authors have to show if their TNF-alpha sample contains the disulfide bond to what extent. To do this, there are several possible methods: thiol quantification using DTNP/DTNB, comparison of reducing and non-reducing SDS-PAGE patterns, MALDI-TOF MS analysis of the trypsin digested fragments, reverse-phase HPLC analysis of S-alkylated proteins in non-reduced and reduced conditions.

Response: We are grateful to your suggestions for improving our manuscript. We did add reducing and non-reducing SDS-PAGE patterns in Figure 8. As TNF-α is a homotrimer, all of the conditions (reducing, non-reducing) just showed 1 band of approximately 21 kDa because, even though the single disulfide bond and subunits are still being cleaved in the reducing conditions, they migrate the same distance in the gel since they are identical.

However, I would like to clear some points regarding the disulfide bond in this cytokine:

[1] Human tumor necrosis factor has single intramolecular disulfide bond between 2 cysteine residues at residues 69 and 101, that was successfully formed many times in E. coli in vivo systems without any problem or need for any cofactor (most recent reports are cited as references [13], [14], [22] in the manuscript). We even have unpublished data for expression of the cytokine in E. coli using three different induction strategies with functional protein yield.   

[2] E. coli shares with eukaryotic cells the reducing cytoplasm that strongly disfavors the formation of stable disulfide bonds in proteins, but do not have structures resembling eukaryotic endoplasmic reticulum. Instead of it, E. coli has an oxidizing periplasm. Therefore, eukaryotic protein expression in bacterial periplasm (especially in engineered competent strains commercially available) is possible. As this cytokine has only a single intramolecular disulfide bond, it is successfully formed in bacteria not only E. coli but also in Streptomyces lividans (reference [16]). The problem appears and the need for cofactors with Multiple Disulfide Bonds as reported in:

de Marco A. Strategies for successful recombinant expression of disulfide bond-dependent proteins in Escherichia coli. Microb Cell Fact. 2009;8:26. Published 2009 May 14. doi:10.1186/1475-2859-8-26

Ma Y, Lee CJ, Park JS. Strategies for Optimizing the Production of Proteins and Peptides with Multiple Disulfide Bonds. Antibiotics (Basel). 2020 Aug 26;9(9):541. doi: 10.3390/antibiotics9090541.

[3] The used system in our study is E. coli chemical protein synthesis machinery not cells, which is the innovation but still the capabilities of living cells are available combined with avoiding negatives of cell confining membranes and possibility of having reducing agent(s) to maintain reduced conditions.

[4] We used E. coli and its cell-free system before in production of another cytokine with a single disulfide bond, human consensus interferon alpha and obtained functional antiviral, anticancer cytokine (references [36], [37]). Note that activity of interferon is dependent on the bond.

[5] E. coli cell-free systems share the same problem with their in vivo counterparts (a limited ability to correctly fold proteins containing multiple disulfide bonds, which is not the case here) as discussed in the following study.

Swartz J. Developing cell-free biology for industrial applications. J Ind Microbiol Biotechnol. 2006 Jul;33(7):476-85. doi: 10.1007/s10295-006-0127-y. Epub 2006 May 9. PMID: 16761165.

[6] The activity of tumor necrosis factor is dependent on the bond (Narachi et al., 1987), thus the activity in cell assays reported in the manuscript as well as other still unpublished bioassays we performed; confirm successful formation of the bond.

Narachi MA, Davis JM, Hsu YR, Arakawa T. Role of single disulfide in recombinant human tumor necrosis factor-alpha. J Biol Chem. 1987 Sep 25;262(27):13107-10. PMID: 3654604.

Comment 2: Recombinant TNF-alpha is commercially available. They should compare the activities of their sample and other product(s).

Response: unfortunately, as this manuscript is derived from PhD thesis as I mentioned in Author contributions section, we do not have resources to repeat assays with commercial control cytokine.  

Comment 3: Figure 8 B: Indicate which peak is the peak of TNF-alpha using an arrow.

Response: Done

Associate Professor Dr Nawal Abd EL-Baky

Round 2

Reviewer 1 Report

I am convinced that the human tumour necrosis factor with an intramolecular disulphide bond can be expressed in the E. coli cell-free expression system. However, the authors should provide a revised version of their manuscript where they address this point clearly. Not all of my concerns have been addressed and those which have been addressed are not provided in a revised version of the manuscript. Basically, my suggestions have been ignored.  

Author Response

Dear Editor in chief

Dear Editor

Dear Reviewers

I would like to express our deep gratitude for all suggestions made by the respected reviewers to improve our work. They helped us a lot to improve the manuscript and make it complete as possible.

Kindly, find below the response to all comments made by the two reviewers in the second round point by point.

Response to reviewer comments:

Reviewer 1:

Comment 1: Iam convinced that the human tumour necrosis factor with an intramolecular disulphide bond can be expressed in the E. coli cell-free expression system. However, the authors should provide a revised version of their manuscript where they address this point clearly.

Response:  The point was clearly addressed in discussion section in the revised version.

Comment  2: You have mentioned that eukaryotic systems are better suited for sTNF-alpha expression due to the possible PTMs with such systems. However, you haven’ t indicate which PTM are needed for sTNF-alpha or if any are needed. Please clarify this point. It seems that in the soluble protein there is a disulfide bond.

Response: This statement was general for TNF products but we removed it to clear any confusion.

Comment  3: Introduce earlier figure 3, it is necessary to understand better the strategy you used to synthesize the gene.

Response: figure 3 illustrates designed oligonucleotides for gene synthesis not the PCR strategy for assembly and amplification. However, considering your concern we added more detailed gene synthesis strategy of PCR assembly and amplification in the results section with Figure 4. Note, the strategy was previously detailed AND DIGRAMMATICALLY ILLUSTRATED in our previous publication [36].

Comment 4: It is not clear while reading the paper which gene you are building into the plasmid for gene expression.

Response: we mention at the beginning that we are going to express sTNF-alpha because we should make it clear for the reader that we express the soluble form not membranous form and presented its exact sequence from data base. Throughout the paper it is named recombinant or cell-free synthesized human TNF-α (this is how it is commonly addressed in literature) and with plasmids the usual way in literature not to write recombinant in the plasmid name so we named it pET-hTNF-alpha for abbreviation. Collectively, all names are true for the protein, recombinant or cell-free or just human TNF-α or sTNF-α. 

Comment 5: Why did you perform the acetone precipitation? How did you obtain the soluble fraction of the protein of interest?

 Response: Detailed in methods section with Both SDS and ELISA.

Comment 6: Omit in the figure legend the following sentence: “which were not modified in 375 any way”. It’s not conceivable that data are modified.

Response: Done

Comment 7: Figure 4 A doesn’t make sense.

Response: Considering your concern we added more detailed gene synthesis strategy of PCR assembly and amplification in the results section with Figure 4. Note, the strategy was previously detailed AND DIGRAMMATICALLY ILLUSTRATED in our previous publication [36].

Comment 8: Figure 4B how do you see that the annealing temperature at 58C produced the best result? This cannot be drawn from the data shown in the figure.

Response: we changed it to slightly improved as the reduction in non-specific bands is not significantly clear in the image but was clear on the gel on transilluminator

Comment 9: Specify if the ELISA assay used to quantify the protein (200 µg/mL) was performed with the protein still in the lysate before purification. However, after CFPS you should centrifuge the lysate and collect the supernatant for ELISA quantification. Why didn’t you perform this step?

Response: the method of obtaining soluble protein was detailed in ELISA method section.

Comment 10: Figure 1 and table 1 should go in the supporting material. Figure 1 is a poor-quality image, please upload an image with better resolution. Please move in the supporting material Figure 2, Figure 3 and table 2. Upload better quality images, especially for Figure 2. Figure 8B should go in the supporting material. Move table 6 in the supporting information. Table 7 in the supporting material. Figure 9 is of poor quality. Table 9 in supporting material. Figure 11 is blurry upload better quality plot.

Response: Unfortunately, we cannot remove all of these essential and important data to supplementary material as this will significantly disrupt the design, value, and flow of the article for us (this a manuscript derived from PhD thesis) and for the readers. The resolution of all images changed after reformatting of the paper by the journal.

Comment 11: Table 3 the soluble fraction is 320.68 µg/mL? You have mentioned that after CPFS you quantified 200 µg/mL through ELISA assay. Clarify this point?

Response: Table 3 the soluble fraction is 320.68 µg/mL: this is total protein content measured by Bradford assay from the CFPS reaction before purification (this mentioned in table as Total protein concentration (µg/ml) and methods and results section). CPFS you quantified 200 µg/mL through ELISA: this is only TNF content from the CFPS reaction before purification. Both detailed in methods.

Comment 12: Figure 9 is of poor quality and the analysis shown is irrelevant for such a simple optimization problem. Remove it from the paper. Report in word that your model has been validated by using residuals, in particular, which kind of residual.

Response: RSM statistical optimization is essential part of the manuscript and PhD thesis from which it was derived, thus cannot be removed. Regarding residual, a residual is a measure of how far away a point is vertically from the regression line. Simply, it is the error between a predicted value and observed one. It is a measure of the strength of the difference between observed and expected values. Standardized residuals greater than 2 (when the ± sign is ignored) are usually considered large.

Comment 13: Explain briefly what a colorimetric MTT assay is. At least the acronym. How did you estimate the IC50 and EC100? Show data plot in the paper.

Response: MTT is the gold standard for determination of cell viability and proliferation since its development by Mosmann in the 1980′s. It is detailed many times in previous studies of others and of our [36, 37]. The equations of estimating IC50 and EC100 starting from absorbance of treated cells compared to control ones, percent of viability, and percent of toxicity as 100-precent of viability, plotting, plot equation and estimation using Graphpad Instat software version 3 are so many details previously described in our work [36, 37] and others. If we detail it again, it will be an unnecessary repeat and plagiarism. But, we added the statement” using 3-(4,5-dimethylthiazol-2-yl)-2,5-diphenyltetrazolium bromide (MTT) dye assay as previously described by Mosmann, 1983 [40] and El-Baky et al., 2011 [37].” in methods. Regarding plots, EC100 against normal cells is calculated from the presented viability plot in manuscript. While IC50 from the same plot but invert viability percent to toxicity percent by equation: percent toxicity= 100-percent viability. In case of IC50 against cancer cells, they are calculated from the plot in manuscript. And EC100 from the reverse plot of viability as mentioned above. The presented plots are the standard for normal and cancer cells MTT results. If we add the reverse plots it will be an addition to length of article and confusion for readers.

Comment 14: The assay depicted in Fig. 13 it’s not convincing, the difference between the two sets of A and B of cell samples

Response: the untreated control cancer cells are living, confluent monolayer forming, green stained cells. On the contrary, treated cells with tumor necrosis factor lost their monolayer, their shape, have fragmented nuclei, stained green, orange, red according to the stage of necrosis, they apparently lost their viability.

Comment 15: show the result of DNA sequencing of the positive clone carrying your gene of interest.

Response: Unfortunately, we did not keep these data for publishing, because we were not asked before in our previous publications [36, 37, 55, cell-free peptide synthesis in Biotechnology Reports 2021] for publishing such data especially as the gene is artificially synthesized not amplified from human genome. We do such analysis in our lab only for confirming the accuracy of PCR gene synthesis strategy.

Comment 16: Cite this paper in the discussion: Microscale to Manufacturing Scale-up of Cell-Free Cytokine Production—A New Approach for Shortening Protein Production Development Timeline Biotechnol Bioeng. 2011 Jul; 108(7): 1570–1578.

Response: Done

Associate Professor Dr Nawal Abd EL-Baky

Reviewer 3 Report

Thank you for your response to my comment. I appreciate your response to [1]-[6] on the disulfide bond formation with references. Since disulfide bonds are an important factor in the structure of TNFα, I recommend adding these descriptions to the Discussion section.

Author Response

Dear Editor in chief

Dear Editor

Dear Reviewers

I would like to express our deep gratitude for all suggestions made by the respected reviewers to improve our work. They helped us a lot to improve the manuscript and make it complete as possible.

Kindly, find below the response to all comments made by the two reviewers in the second round point by point.

Response to reviewer comments:

Reviewer 2:

Comment:  Since disulfide bonds are an important factor in the structure of TNFα, I recommend adding these descriptions to the Discussion section.

Response: thank you for your suggestion, this point was clearly addressed in discussion section in the revised version.

Associate Professor Dr Nawal Abd EL-Baky